# The impact of elevation and population density on dengue incidence and force of infection across the Philippines: Implications for climate-adapted surveillance

Ava Kristy Sy[1,2‡], Joseph Biggs[3‡*], Shahida K. Chowdhury[3], Mary Ann Quinones[1], Ferdinand V. Salazar[4], William Jones-Warner[5], James Ashall[5], Ma. Nemia L. Sucaldito[6], Eva Cutiongco-de la Paz[7], Maria Rosario Z. Capeding[8], Carmencita D. Padilla[7], Sharon Y. A. M. Villanueva[2], Martin L. Hibberd[5,7], Julius Clemence R. Hafalla[5]

**1** Department of Virology, Research Institute for Tropical Medicine, Manila, Philippines, **2** Department of Medical Microbiology, College of Public Health, University of the Philippines, Manila, Philippines, **3** International Statistics and Epidemiology Group, Department of Infectious Disease Epidemiology and International Health, London School of Hygiene and Tropical Medicine, London, United Kingdom, **4** Department of Medical Entomology, Research Institute for Tropical Medicine, Manila, Philippines, **5** Department of Infection Biology, London School of Hygiene and Tropical Medicine, London, United Kingdom, **6** Philippine Epidemiology Bureau, Department of Health, Manila, Philippines, **7** Institute of Human Genetics, National Institute of Health, University of the Philippines, Manila, Philippines, **8** Clinical Trial Unit, Tropical Disease Foundation Inc., Makati, Philippines

‡ Co-primary authors.
* Joseph.biggs1@lshtm.ac.uk

## Abstract

Climate change is accelerating the geographical expansion of Aedes mosquito vectors, facilitating the emergence of arboviral diseases such as dengue in new regions. However, there is limited understanding of how environmental factors, such as temperature and population density, differentially influence key metrics of dengue transmission. This study investigates how elevation (a proxy for temperature) and population density affect dengue incidence and force of infection (FOI), both independently and dependently, across the Philippines, with implications for climate-adapted surveillance and control. We conducted a nationwide, cross-sectional survey across the Philippines (2013–2019), combining national dengue case data (N = 1,112,317) with antibody IgG seroprevalence data from a representative sample (N = 22,270). Dengue FOI (the predicted rate at which individuals become exposed to DENV annually) and incidence were estimated across elevation and population density strata using catalytic models. Regression analyses were used to assess interactions between geographic factors and the dengue FOI. Elevation and population density were associated with FOI, but not with reported dengue incidence, which varied unpredictably over space and time. Urban, low-elevation barangays had the highest FOI (13.2%), while rural and urban, high-elevation areas had lower FOI (<7.7%). These patterns held across years, island groups, and regions. A statistical interaction between elevation and population density on the dengue FOI improved

**Data availability statement:** The data underlying this study were collected as part of routine surveillance activities and are subject to ethical and governance restrictions, and the conditions under which these data were collected did not include provision for public data sharing. Case report surveillance data are held by the Epidemiology Bureau, Department of Health Philippines, and serological surveillance data are held by the Research Institute for Tropical Medicine (RITM); these data are not publicly available. Researchers who meet the criteria for access to confidential data may submit requests to the respective institutions (Department of Health Philippines: dohosec@doh.gov.ph; Research Institute for Tropical Medicine: phu@ritm.gov.ph), and access is subject to institutional approval and data governance requirements.

**Funding:** J.C.R.H. was supported by the The Royal Society (grant CHG\R1\170061). M.L.H. and C.D.P. were supported by the Newton Fund Institutional Links programme (project number 216416089), delivered by the British Council and the Commission on Higher Education. The funders had no role in study design, data collection and analysis, decision to publish, or preparation of the manuscript.

**Competing interests:** The authors have declared that no competing interests exist.

model fit (p < 0.0053) and revealed population density is a driver of transmission intensity in low lying areas (<1200m) but not in high elevation areas (>1200m). Dengue burden in the Philippines is highest in low-lying, urban areas and significantly lower in high-elevation zones, irrespective of population density, where cooler temperatures prevail. FOI is a more reliable metric than incidence for understanding transmission dynamics and guiding interventions. Our findings highlight the need for geographically tailored control strategies, particularly as climate change alters environmental and demographic conditions.

## Author summary

Dengue is spreading into new areas as temperatures rise and cities expand, but reported case numbers do not always reflect where transmission is truly occurring. In this study, we examined how elevation (a marker of temperature) and population density influence dengue transmission across the Philippines. We analysed over one million reported dengue cases alongside serological data from more than 22,000 patients to estimate both dengue incidence and the *force of infection*—the underlying rate at which people are exposed to dengue, including infections without noticeable symptoms. Reported dengue incidence varied unpredictably across regions and years and showed no consistent relationship with elevation or population density. In contrast, force of infection revealed clear and stable patterns. Transmission was highest in low-lying, densely populated urban areas and substantially lower at higher elevations, regardless of population density. Population density increased transmission only in low-elevation areas, where warmer temperatures favour mosquito survival. These findings show that commonly used epidemiological metrics including case counts and incidence, based on reported cases, can misrepresent dengue risk. Measures that capture underlying transmission provide more reliable guidance for surveillance and control. As climate change warms higher-elevation regions, geographically targeted strategies will be essential to prevent future dengue outbreaks.

## Introduction

Climate change is reshaping global health risks by accelerating the spread of vector-borne diseases such as dengue [1]. In 2024, a record 14 million dengue cases and approximately 9,000 dengue-related deaths were reported globally, representing a 12-fold increase in reporting since 2014 [2]. Dengue virus (DENV) is primarily transmitted by *Aedes* mosquitoes, particularly *Ae. aegypti* and *Ae. albopictus*, which thrive in warm, humid environments and are well adapted to urban settings. Rising temperatures, shifting rainfall patterns, and rapid urbanisation have expanded their geographical range, leading to outbreaks in previously unaffected regions, including Southern Europe [3]. Understanding of how environmental factors shape dengue transmission remains limited, largely due to underreporting caused by inconsistent

healthcare access, misdiagnosis, and the high prevalence of asymptomatic infections [4]. Unpacking how these geographical factors influence both reported incidence and the underlying force of infection (FOI) is crucial for guiding climate-resilient arbovirus control and surveillance strategies.

*Aedes aegypti* and *Aedes albopictus* are highly anthropophilic vectors with limited flight ranges of approximately 100–250 metres and short life spans of only a few weeks [5,6]. High human population densities facilitate sustained dengue transmission between vectors and human hosts. Urban environments with abundant standing water in artificial containers, gutters, and storm drains, provide ideal breeding sites for *Aedes* mosquitoes [7,8]. Air temperature also plays a critical role in influencing their abundance, survival, and vectorial capacity. The optimal temperature for maximum vectorial capacity is estimated at around 29 °C, though this may vary with daily fluctuations in temperature [9]. In cooler, temperate conditions, the *Aedes* gonotrophic and reproductive cycles slow, and the extrinsic incubation period lengthens, reducing the number of infectious adult females capable of transmitting dengue virus [10]. These environmental and socio-economic factors contribute to the hyperendemic nature of dengue in many tropical urban centres worldwide.

Numerous studies have investigated the influence of elevation and population density on reported dengue incidence. In Nepal [11], Indonesia [12], and Pakistan [13], lower incidence was observed in higher-elevation regions with cooler climates. Moreover, studies in Vietnam [14], Taiwan [15], and Pakistan [13] found lower dengue incidence in areas with lower population density. However, other research found no clear relationship between dengue incidence and temperature or population density [12,16]. These inconsistencies have been attributed to factors such as socio-economic differences, healthcare seeking behaviour, and variable healthcare access.

Other studies have explored how elevation and population density influence alternative metrics of dengue transmission, such as FOI, derived from age-seroprevalence surveys. FOI measures serological exposure to DENV regardless of symptoms, making it a more reliable indicator of true underlying transmission intensity as it is unaffected by variability in reporting and care-seeking [17]. In Colombia, seroprevalence surveys revealed higher FOI estimates in urban compared to rural areas [18], while a survey in Malaysia showed exposure to DENV declined with increasing elevation [19]. A multi-country study assessed both incidence and FOI in relation to elevation and population density, finding associations with FOI but not with incidence [20]. The authors suggested this mismatch may reflect inconsistent case definitions and the aggregation of incidence data across large administrative areas. Standardised data at finer spatial scales may therefore better capture these relationships. Additionally, interactions between ecological factors may explain the divergence between FOI and incidence. For example, transmission may be higher in low-elevation regions, but underreporting may occur in rural areas where healthcare access is limited.

In this study, we examined the effects of elevation and population density on dengue incidence and FOI, both individually and in combination. Using routinely collected case surveillance data and serological data from reported cases, we estimated incidence and FOI across multiple geographical strata. Understanding how these factors influence different measures of dengue transmission is essential for evaluating the suitability of these metrics for future control and monitoring programmes. By highlighting the added value of integrating climate-adapted measures into dengue surveillance systems, this approach may enhance outbreak prediction, improve resource allocation, and inform policies to mitigate the impact of climate change on dengue transmission.

## Methods

### Ethics statement

Ethical approval for the study was obtained from the institutional review boards of the Research Institute for Tropical Medicine, Philippines (2017–2014) and the London School of Hygiene and Tropical Medicine, UK (17965). Written formal consent was obtained from study participants prior to enrolment. For those under 18 years, assent and written informed

consent from parents/guardians was required prior to enrolment. Participants also provided permission for the use of sera for research purposes. All unique identifiers were removed prior to data acquisition.

## Philippine dengue surveillance and data collection

Dengue is a notifiable disease in the Philippines. The country is divided into 3 main island groups: Luzon, Visayas, and Mindanao, which are further divided into 17 regions, 1,488 municipalities and 42,046 barangays (Fig 1). Suspected dengue cases are reported through Disease Reporting Units (DRUs), which include both small health facilities and major hospitals nationwide. Cases are classified according to the WHO criteria as dengue without warning signs, dengue with warning signs, or severe dengue. Dengue with warning signs is a symptomatic dengue infection accompanied by clinical indicators, such as abdominal pain, persistent vomiting, fluid accumulation, mucosal bleeding, lethargy, and/or liver enlargement [21]. Basic epidemiological data, including age, sex, and reporting location, were also collected for reported cases. This information is centrally compiled by the Philippine Epidemiology Bureau. In parallel, laboratory-based dengue surveillance is conducted by the Research Institute for Tropical Medicine. Serum samples are obtained from a random subset of dengue patients presenting to sentinel DRUs and subjected to laboratory testing. Detailed dengue surveillance operations in the Philippines have been described previously [17].

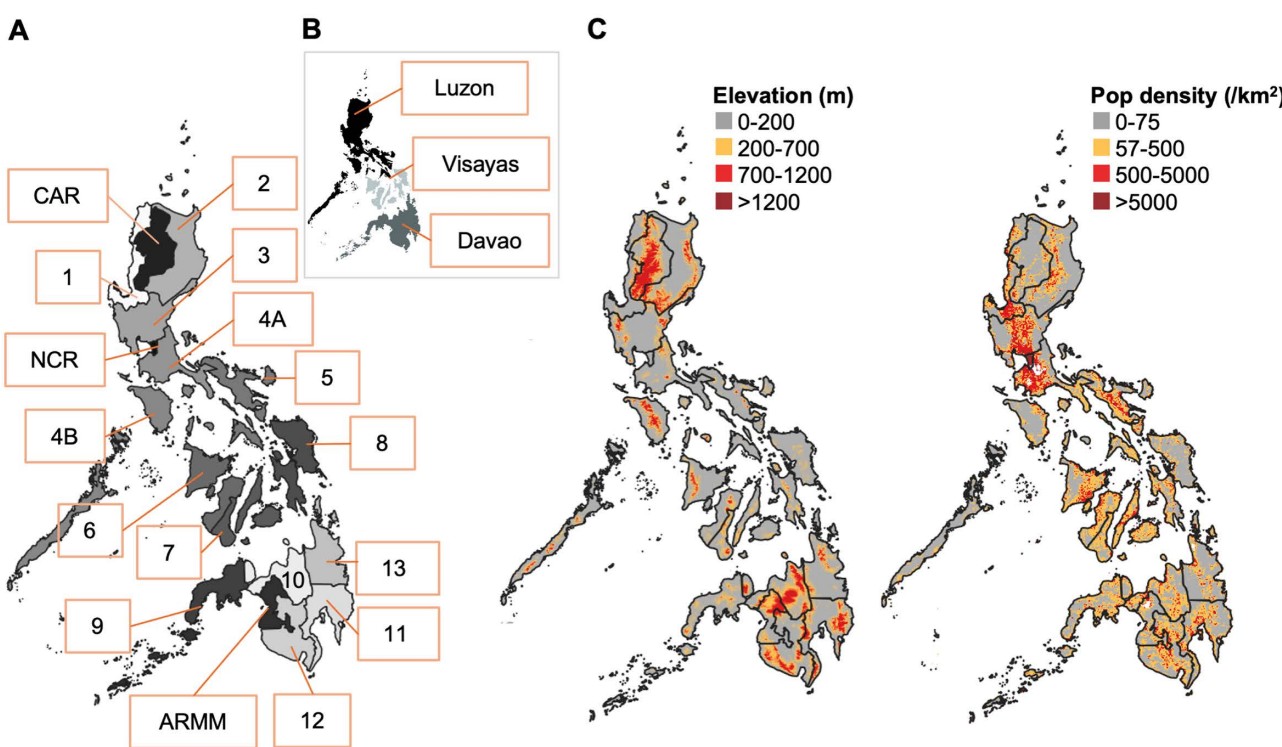

**Fig 1. Study Location. A**: Regions of the Philippines. CAR: Cordillera Administrative Region; NCR: National Capital Region; ARMM: Autonomous Region in Muslim Mindanao. **B**: Major island groups of the Philippines. **C**: Average elevation (metres) and population density (individuals/km²) per barangay across the Philippines. The country is divided into three main island groups: Luzon, Visayas and Mindanao, further divided into 17 regions, 1,488 municipalities and 42,046 barangays. Base map shapefile (Philippines administrative boundaries) obtained from the Humanitarian Data Exchange (HDX), provided by the United Nations Office for the Coordination of Humanitarian Affairs (OCHA): https://data.humdata.org/dataset/cod-ab-phl. Data are available under an open license compatible with CC BY.

In this cross-sectional study, we obtained data from patients with reported dengue across the Philippines between 2013 and 2018. In addition to these dengue case reports, we analysed laboratory data from those who provided serum samples between 2014–2019. Previous work has shown similar demographic characteristics of these related datasets supporting the representativeness of the surveyed case report data [17]. Geographical data were obtained at the local barangay level. A 2015, 100-metre resolution Digital Elevation Model (USGS Earth Explorer; USA) was used to estimate the average barangay elevation (metres) in ArcGIS (v10.5). In the Philippines, the association between decreasing temperature and increasing elevation has been well documented [22]. Population data from the 2015 Philippine Census and annual population growth rates from 2010 to 2015 (Philippine Statistics Authority) were used to estimate barangay-level population for each year of the study period (2013–2019). Population density was calculated as the annual barangay population divided by the barangay area ($km^2$). Daily maximum and minimum air temperature (°C) and elevation (metres) data from 55 weather stations across the Philippines, collected between 2010 and 2015 (PAGASA: Philippine Atmospheric, Geophysical and Astronomical Services Administration; Manila; Philippines), were also utilised.

## Laboratory analysis

Serum samples collected from dengue cases were subjected to molecular and serological analysis as described previously in [17]. A fourplex real-time polymerase chain reaction (RT-PCR) assay was used to detect DENV1–4 RNA in serum samples. Individuals with threshold cycle (Ct) values <36 were classified as PCR-positive. Samples were also tested for anti-DENV IgM and IgG using Panbio capture ELISAs, generating standardised antibody Panbio units (Cat No: 01PE10/01PE20, Alere, Brisbane, Australia). IgG and IgM seroprevalence thresholds representing serological evidence of dengue exposure were >2.2 and >9.9 Panbio units, respectively.

## Statistical analysis

Dengue incidence was estimated across barangay-level elevation and population density strata. In each stratum, incidence was calculated as the number of reported cases per total population per year, multiplied by 1,000. Dengue FOI estimates across identical elevation and population density strata were generated using two previously described methods [17]. First, reversible catalytic models were fitted using maximum likelihood to calculate age-IgG seroprevalence and seroconversion rates among those without active dengue infections (PCR- and IgM-) using the *revcat* command in STATA (v17) (Eq 1). Those with active dengue infections (PCR+ or IgM+) were excluded, as IgG levels are influenced by primary and post-primary immune status and disease progression. Assuming equal contributions of circulating serotypes over time and the waning of IgG to undetectable levels [17], the probability of being IgG seropositive at a given age (a) was modelled by fitting seroconversion ($\lambda$) and seroreversion ($\rho$) parameters by least squares:

$$P(a) = \frac{\lambda}{\lambda + \rho} \left[ 1 - e^{-(\lambda + \rho)a} \right]$$

(1)

Seroconversion estimates were subsequently used to calculate FOI (the annual proportion exposed to DENV (Eq 2):

$$FOI = 1 - e^{-(\lambda)}$$

(2)

Second, FOI was also estimated based on the mean age (a) of cases reporting with WHO-classified dengue warning signs as previously described [17] (Eq 3):

$$FOI = 0.023 + 0.630 \times 0.894^{(a)}$$

(3)

Univariate and multivariate logistic regression models were utilised to generate unadjusted and adjusted odds ratios (ORs) for being DENV IgG+ across elevation and population density strata using the *xtlogit* command in STATA (v17).

Explanatory variables included elevation and population density, with adjusted models incorporating age, sex, island group, and year. Both models include for random effects at the barangay level to account for correlations at low spatial scales. Active dengue infections (PCR+ or IgM+) were excluded to avoid confounding effects of immune status and disease progression on IgG levels of reporting patients.

Mixed-effects linear regression models were employed to examine the impact of elevation and population density, with and without an interaction, on the barangay-level DENV FOI estimated according to the age of case reports with dengue warning signs using the *mixed* command in STATA (v.17). Models were adjusted for study year and island group and accounted for random effects at the municipality level (the administrative level above barangay). The inclusion of an interactive term between barangay population density and elevation was determined based on superior model fit (likelihood ratio test (LRT), $p < 0.05$).

## Results

A total of 1,112,317 dengue case reports from 2013–2018, along with 22,270 serologically surveyed case reports from 2014–2019, covering the entire Philippines, were included in this study. Approximately 40% of case reports were 6–15 years old, 56% reported on day 3–4 of disease onset, and 52% presented with dengue warning signs. Dengue cases were predominantly reported in urban, low-lying areas, which is where most of the population resides (Fig 1) (S1 Table). Of the collated reports, 85.4% (949,796/1,112,317) originated from barangays with an average elevation of <200 metres, and 78.4% (871,778/1,112,317) were from barangays with a population density of >500/km². However, in the Cordillera Administrative Region (CAR), Autonomous Region in Muslim Mindanao (ARMM), and Region 10, dengue cases were also reported from higher-elevation and densely populated areas (S1 Fig). In the CAR region, 38.0% (15,680/41,250) of dengue cases were from urban barangays (>500 km2) located above 200 meters. The collated and surveyed dengue case reports included in this study exhibited similar distributions, as they were derived from comparable geographic strata across similar regions (S1 Fig). Lastly, lower daily maximum and minimum air temperatures (2010–2015 data) were observed in areas with higher elevations (S2 Fig).

### The individual impact of elevation and population density on dengue incidence and force of infection

We estimated dengue incidence across different elevation and population density strata. Across the Philippines (Fig 2A), by major island group (Fig 2B), and by year (Fig 2C), no clear trend was observed between elevation and dengue incidence. In different island groups, dengue incidence either increased or decreased with elevation. For instance, in Luzon, incidence increased from 1.63 cases/1,000/year at 0–200 metres to 3.48 cases/1,000/year at 1,200–1,700 metres. Conversely, in Mindanao, incidence decreased from 1.65 cases/1,000/year at 0–200 metres to 0.69 cases/1,000/year at 1500–1700 metres. Notably, higher incidence observed in elevated areas was temporally restricted to years 2014, 2016 and 2017.

In contrast, a modest relationship was observed between population density and dengue incidence. Incidence increased slightly with population density, from 1.06 cases/1,000/year in barangays averaging <75 individuals/km² to 1.88 cases/1,000/year in barangays averaging >5,000 individuals/km² (Fig 2A). This trend persisted when stratified by island group and year, although exceptions were noted in Luzon and certain years (Fig 2B and 2C). Overall, these findings suggest that dengue incidence trends are spatially and temporally sporadic across the Philippines, with no consistent association with either elevation or population density.

We next explored the impact of elevation and population density on the probability of having prior serological exposure (IgG+) to DENV across the Philippines from surveyed case reports (2014–19). The odds of being exposed to dengue increased with decreasing elevation and increasing population density (Fig 2D). Compared to individuals residing at 1,200–1,700 metres, those at 0–200 meters were significantly more likely to be DENV IgG-positive (aOR: 5.92 [95%CI: 3.87-9.04], $p < 0.001$). Similarly, individuals from barangays with >5,000/km² population density were more likely to be

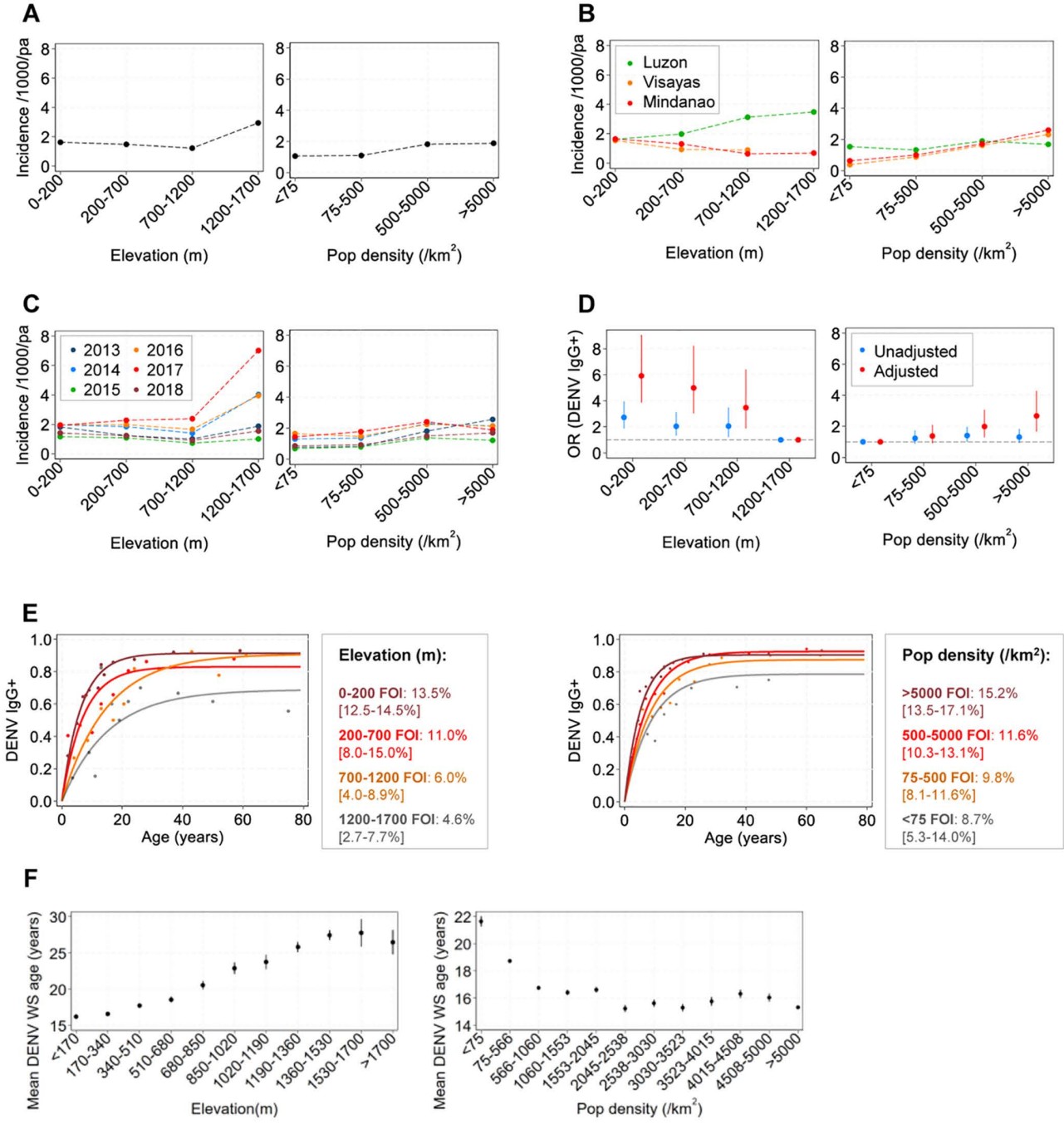

**Fig 2. The impact of elevation and population density on dengue transmission dynamics. A-C**: Dengue incidence by elevation and population density strata across the Philippines **(A)**, island groups **(B)**, and study years **(C)**. **D**: The crude and adjusted odds ratios of being DENV IgG+ by elevation and population density strata. Multivariate models, adjusted for age, region, year, elevation/population density and random effects at the barangay level achieved superior model fit to univariate models according to lower AIC. **E**: Dengue age-seroprevalence and FOI by elevation and population density strata according to catalytic models. Dots: observed age-seroprevalence, lines: fitted age seroprevalence, brackets: FOI 95%CIs. **F**: Mean age of patients reporting with dengue warning signs (WS) by elevation and population strata. Error bars: 95%CIs.

IgG-positive compared to those from lower population density barangays (0–75/km²) (aOR: 2.67 [95%CI: 1.67-4.28], p<0.001). After adjusting models for age, region, year, elevation or population density, the association between increased IgG exposure, decreasing elevation, and increasing population density became statistically significant (non-overlapping 95%CIs with 1) and achieved superior model fits to univariate models (lower AIC values).

By further stratifying DENV IgG exposure by age, we estimated the dengue FOI by elevation and population density strata across the Philippines using catalytic models (Fig 2E) (S3 Fig). As barangay elevation increased, the DENV FOI decreased from 13.5% [95%CI: 12.5-14.5%] at elevations between 0-200m to 4.6% [95%CI: 2.7-7.7%] at elevations >1,200m. Conversely, as population density increased, the dengue FOI increased from 8.7% [95%CI: 5.3%-14.0%] in areas with 0–75 individuals/km$^2$ to 15.2% [95%CI:13.5-17.1%] in barangays with >5000 individuals/km$^2$.

To further characterise the relationship between FOI, elevation, and population density, we estimated the FOI using an alternative method based on the age at which individuals reported with dengue warning signs. Across more granular elevation and population density strata, patients reported at younger mean ages in lower-elevation and high population-density areas (Fig 2F). This was expected, as younger reporting ages suggest a higher FOI. Collectively, these results indicate that dengue transmission is more prominent in low-lying areas and more urban areas.

To investigate whether the dengue FOI could be accurately estimated using the age of patients with warning signs, we compared these estimates to gold-standard catalytic model FOI estimates across elevation and population density categories (Table 1). Despite small discrepancies between the two methods, the overall estimates aligned well, further supporting the use of the age of patients with warning signs as a suitable surrogate indicator of transmission intensity. It should be noted, however, that estimates were derived from separate datasets collected during slightly different time periods.

## The combined impact of population density and elevation on dengue incidence and force of infection

We examined whether the effects of elevation and population density on dengue incidence and FOI were dependent or independent of each other. When stratifying dengue incidence by both population density and elevation, spatio-temporal sporadic dengue case reporting was still observed across the Philippines, major island groups, selected regions, and

**Table 1. Dengue FOI estimates across elevation and population density strata according to catalytic models and the age of patients reporting with dengue warning signs. Estimates from catalytic models utilised surveyed case report data with laboratory metrics collected between 2014 & 2019. FOI estimates were obtained from all collated case reports between 2013 & 2018. DENV with WS: dengue with warning signs.**

| | FOI | | FOI | |
| --- | --- | --- | --- | --- |
| | Catalytic model[a] | | Age of dengue case[b] | |
| | % | [95%CI] | % | [95% CI] |
| **Population density (/km²)** | | | | |
| >5000 | 15.20 | [13.5-17.1] | 12.73 | [12.6-12.8] |
| 500-5000 | 11.57 | [10.3-13.1] | 11.73 | [11.7-11.8] |
| 75-500 | 9.79 | [8.1-11.6] | 9.42 | [9.3-9.5] |
| 0-75 | 8.70 | [5.3-14.0] | 7.58 | [7.4-7.8] |
| **Elevation (metres)** | | | | |
| 0-200 | 13.50 | [12.5-14.5] | 11.78 | [11.7-11.8] |
| 200-700 | 11.04 | [8.0-15.0] | 10.71 | [10.6-10.9] |
| 700-1200 | 6.01 | [4.0-8.9] | 7.38 | [7.1-7.6] |
| 1200-1700 | 4.59 | [2.7-7.7] | 5.30 | [5.1-5.5] |

[a]DENV FOI according to age-seroprevalence from a catalytic model.

[b]DENV FOI according to the mean age of dengue case reports with warning signs.

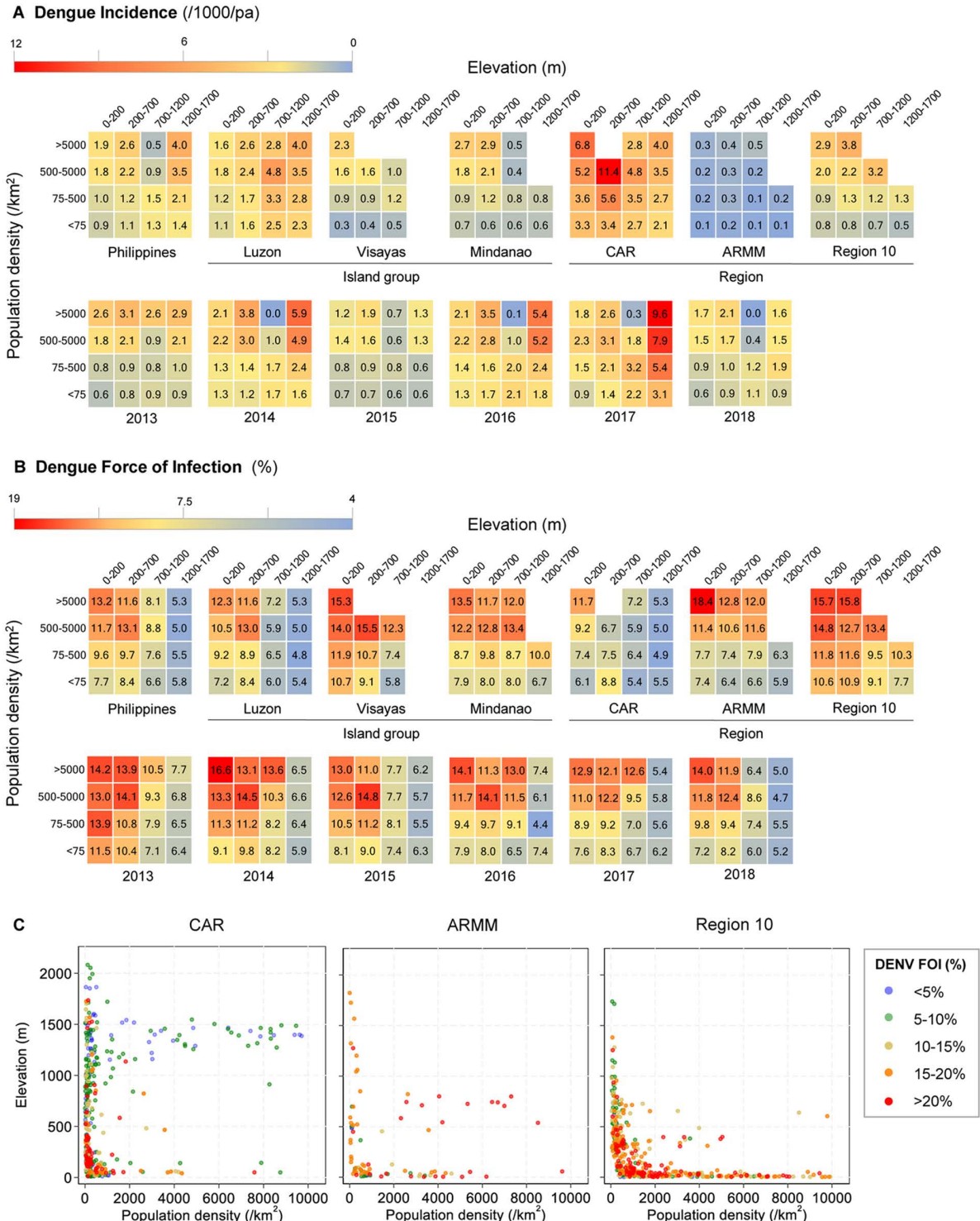

**Fig 3. The combined impact of elevation and population density on dengue reported incidence and DENV FOI.** Reported dengue incidence (cases/1000/annum) (**A**) and DENV FOI (**B**) stratified by both population density and ground elevation across the entire Philippines, by major island group, by selected region and year. **C**: Barangay-level DENV FOI% by population density and elevation for selected regions aggregated over the study period (2013-2018). Only includes barangays where >20 cases reported. FOI: predicted annual rate at which individuals become exposed to DENV based on the age at which individuals with DENV warning signs report.

study years (Fig 3A). Notably for specific years, there was a marked increase in incidence in high-elevation, densely populated regions, such as CAR, while remaining consistently low overall in ARMM.

In contrast, following stratification by both elevation and population density strata, DENV FOI was highest in low-elevation and densely populated areas (Fig 3B). Across all low-elevation (0-200m) and densely populated (>5000/km$^2$) barangays, the average FOI was 13.2% [95%CI: 13.1-13.3%]. This significantly exceeded FOI estimates in both elevated, urban areas (1200-1700m; >5000/km$^2$) and low-elevation, rural zones (0-200m; <75/km$^2$), with corresponding FOI estimates of 5.3% [95%CI: 5.1-5.5%] and 7.7% [95%CI: 7.2-8.1%], respectively. This trend was still observed following stratification by island group, selected regions, and study years. We further explored this associated at lower administrative levels (barangays) (Fig 3C). Within defined geographical strata, we observed substantial heterogeneity in DENV FOI at the barangay level. Therefore, despite demonstrating low-lying urban centres experience higher dengue burden overall, at lower spatial scales, dengue burden is highly variable.

Finally, upon estimating the barangay-level FOI based on the age of those with dengue warning signs, we used a mixed effect linear regression model to further characterise the interaction between population density and elevation on dengue burden. Firstly, inclusion of an interactive term improved model fit (LRT, p = 0.005) providing further evidence that the effects of population density and elevation on DENV FOI intensity are interdependent. Secondly, using this model to predict the DENV FOI across geographical strata showed in lower elevation barangays (<1200m), the DENV FOI increased with increasing population density while in high elevation areas (>1200m), no trend was observed between population density and the DENV FOI (Fig 4).

## Discussion

As global warming intensifies, predicting its impact on vector-borne diseases such as dengue becomes increasingly critical. In this nationally-representative study across the Philippines, we examined the independent and synergistic effects of elevation and population density on reported dengue incidence and DENV FOI at fine spatial scales. Results demonstrate

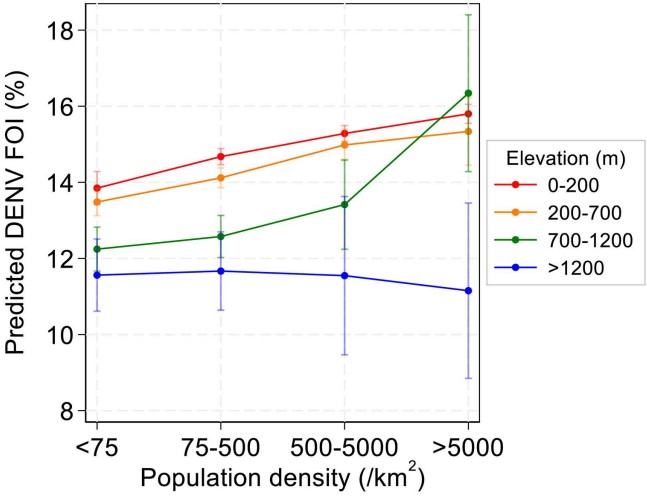

**Fig 4. The effects of population density and elevation on the predicted barangay-level DENV FOI according to the age of reported cases with dengue warning signs.** FOI was predicted using a mixed effects linear regression model adjusted for island group, study year and accounted for random effects at the municipality level. The inclusion of an interactive term between population density and elevation improved model fit (LRT, p = 0.0053). Error bars: 95%CIs.

underlying dengue transmission was highest in low lying urban areas whereas passive incidence was spatially and temporally sporadic during the six-year study period. This key mismatch illustrates the challenges of characterising dengue transmission dynamics based on case surveillance data alone and reinforces the value of utilising alternative metrics of dengue transmission intensity to allocate limited control intervention resources, monitor changes in transmission and identify areas at risk of future outbreaks.

Our findings are consistent with studies from Malaysia [19], Colombia [18], and Bangladesh [23] showing higher dengue FOI in urban areas and lower transmission at higher elevations. By using nationally representative, fine-scale data, our study is one of the first to jointly analyse elevation and population density at scale across an entire country. Results highlight that despite being highly heterogenous at fine spatial scales, overall dengue transmission intensity is highest in low-lying urban centres, and population density is less of a driver of dengue transmission in high-elevation areas. The latter likely reflects cooler temperatures limiting the ability of *Ae. aegypti* and *Ae. albopictus* to proliferate and efficiently transmit dengue, regardless of local human population density [9,24].

Notably, we found no consistent association between incidence and either elevation or population density, contrary to some previous reports [11–14]. Since the FOI and incidence were derived from related datasets using standardised WHO case definitions, this discrepancy is unlikely to stem from diagnostic inconsistencies. Instead, it may reflect differences in health-seeking behaviour and reporting infrastructure. For example, ARMM region exhibited low reported incidence despite a relatively high FOI, possibly due to data-sharing limitations under its autonomous governance [25]. Conversely, the CAR region showed elevated case reporting despite lower FOI, which may be explained by increased health-seeking behaviour following increased awareness campaigns and/or concern associated with unfamiliar dengue symptoms [26]. These regional contrasts highlight the limitations of relying on incidence alone and underscore the importance of alternative measures for accurately capturing dengue transmission dynamics.

This study highlights the importance of integrating various measures of dengue transmission intensity into climate-adapted dengue surveillance and control practises. For instance, low lying urban centres in dengue endemic countries should be earmarked for monitoring and surveillance, irrespective of the number of case reports, as they remain at risk of future outbreaks. As suggested by the WHO, a range of transmission intensity metrics should ideally be collected for effective surveillance [27]. However, additional serological and entomological monitoring is costly and logistically challenging to implement. Tracking the average age at which patients present with dengue symptoms offers a more practical alternative. This metric correlates well with gold-standard force of infection estimates across different elevations and population density levels and has been validated in urban settings [17]. Nevertheless, further studies are required to confirm whether these estimates reliably reflect transmission intensity at finer spatial and temporal scales before wider adoption.

In our study, we report how daily minimum and maximum temperatures decrease with increasing elevation and heavily impact dengue transmission intensity. As climate change raises global temperatures and alters weather patterns, high elevation, urban areas may face an increasing risk of dengue outbreaks, a trend potentially already emerging as studies reporting that formerly cooler elevated regions are now becoming suitable habitats for *Aedes* mosquitoes [28]. Based our estimates of minimum and maximum average temperatures, high elevation region temperatures only need to increase by approximately 5 degrees Celsius to accommodate dengue transmission. Consequently, monitoring changing temperature patterns in such regions remains a key priority for arbovirus control.

There are some limitations in this study. First, elevation was utilised as a proxy for temperature. Although ground elevation was shown to correlate with weather station air temperature, fixed elevation measures cannot account for intra- and inter-annual fluctuations in air temperature, which likely impact transmission dynamics. Second, barangay populations were estimated using past population growth trends, assuming only gradual changes in population over time. It does not account for sudden or unexpected changes in population that can ensue with sharp economic changes, rapid urbanisation, and mass migration. Third, the impact of population density on dengue transmission is likely confounded by

socioeconomic status, a factor that we could not incorporate in this analysis. Lastly, as described previously [17], FOI esti-mates from case reports using catalytic models may be slightly overestimated. These estimates were based on non-active dengue infections among individuals seeking care, a subset of the population potentially more inclined to report due to previous dengue exposure.

## Conclusion

These findings underscore the importance of integrating FOI measures into surveillance systems to enhance outbreak detection and response, particularly as global warming increases dengue risk in regions previously considered cooler. Climate adaptation strategies, such as improving vector surveillance, strengthening health infrastructure, and implement-ing proactive vector control measures in vulnerable areas will be critical to mitigating the public health impacts of climate change on dengue transmission.

## Supporting information

**S1 Table. The total population and number of collated and surveyed dengue case reports across the Philippines during the study period.**
(DOCX)

**S1 Fig. The average population densities (/km$^2$) and elevations (metres) of the collated (A) and surveyed (B) den-gue case reports between 2013–2018 and 2014–2019 by Philippine regions, respectively.**
(DOCX)

**S2 Fig. The association between weather station ground elevation (metres) and averaged daily maximum/mini-mum air temperature between 2010–15.**
(DOCX)

**S3 Fig. Dengue age-seroprevalence and FOI by elevation and population density strata according to catalytic models.**
(DOCX)

## Acknowledgments

We wish to thank all the study personnel across all disease reporting units for collecting data from patients. We also thank participating staff at the Philippine Epidemiology Bureau, Research Institute for Tropical Medicine and the UP Office of International Linkages Continuous Operational and Outcomes-Based Partnership for Excellence in Research and Aca-demic Training Enhancement (COOPERATE) program

## Author contributions

**Conceptualization:** Ava Kristy Sy, Joseph Biggs, Martin L. Hibberd, Julius Clemence R. Hafalla.

**Data curation:** Ava Kristy Sy, Mary Ann Quinones, William Jones-Warner, James Ashall, Ma. Nemia L. Sucaldito.

**Formal analysis:** Ava Kristy Sy, Joseph Biggs, Shahida K. Chowdhury, Martin L. Hibberd.

**Funding acquisition:** Martin L. Hibberd, Julius Clemence R. Hafalla.

**Investigation:** Ferdinand V. Salazar, Carmencita D. Padilla, Sharon Y. A. M. Villanueva, Martin L. Hibberd, Julius Clemence R. Hafalla.

**Methodology:** Ava Kristy Sy, Joseph Biggs, William Jones-Warner, James Ashall, Ma. Nemia L. Sucaldito, Eva Cutiongco-de la Paz, Maria Rosario Z. Capeding, Carmencita D. Padilla, Sharon Y. A. M. Villanueva, Martin L. Hibberd, Julius Clemence R. Hafalla.

**Project administration:** Ava Kristy Sy, Ferdinand V. Salazar, Maria Rosario Z. Capeding, Martin L. Hibberd.

**Resources:** Ava Kristy Sy, Ma. Nemia L. Sucaldito, Sharon Y. A. M. Villanueva.

**Supervision:** Ava Kristy Sy, Ferdinand V. Salazar, Eva Cutiongco-de la Paz, Martin L. Hibberd, Julius Clemence R. Hafalla.

**Validation:** Ava Kristy Sy, Martin L. Hibberd, Julius Clemence R. Hafalla.

**Visualization:** Joseph Biggs, Shahida K. Chowdhury, Julius Clemence R. Hafalla.

**Writing – original draft:** Ava Kristy Sy, Joseph Biggs, Martin L. Hibberd, Julius Clemence R. Hafalla.

**Writing – review & editing:** Ava Kristy Sy, Joseph Biggs, Shahida K. Chowdhury, Mary Ann Quinones, Ferdinand V. Salazar, William Jones-Warner, James Ashall, Ma. Nemia L. Sucaldito, Eva Cutiongco-de la Paz, Maria Rosario Z. Capeding, Carmencita D. Padilla, Sharon Y. A. M. Villanueva, Martin L. Hibberd, Julius Clemence R. Hafalla.

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
