## [Decision Letter · Decision Letter 0]

17 Mar 2026

PNTD-D-26-00215The impact of altitude and population density on dengue incidence and force of infection across the Philippines: Implications for climate-adapted surveillancePLOS Neglected Tropical Diseases Dear Dr. Biggs, Thank you for submitting your manuscript to PLOS Neglected Tropical Diseases. After careful consideration, we feel that it has merit but does not fully meet PLOS Neglected Tropical Diseases's publication criteria as it currently stands. Therefore, we invite you to submit a revised version of the manuscript that addresses the points raised during the review process. Please submit your revised manuscript by May 16 2026 11:59PM. If you will need more time than this to complete your revisions, please reply to this message or contact the journal office at plosntds@plos.org.  Please include the following items when submitting your revised manuscript:* A letter that responds to each point raised by the editor and reviewer(s). You should upload this letter as a separate file labeled 'Response to Reviewers'. This file does not need to include responses to any formatting updates and technical items listed in the 'Journal Requirements' section below.* A marked-up copy of your manuscript that highlights changes made to the original version. You should upload this as a separate file labeled 'Revised Manuscript with Track Changes'.* An unmarked version of your revised paper without tracked changes. You should upload this as a separate file labeled 'Manuscript'. If you would like to make changes to your financial disclosure, competing interests statement, or data availability statement, please make these updates within the submission form at the time of resubmission. Guidelines for resubmitting your figure files are available below the reviewer comments at the end of this letter. We look forward to receiving your revised manuscript. Kind regards, Nikos VasilakisSection EditorPLOS Neglected Tropical Diseases Nikos VasilakisSection EditorPLOS Neglected Tropical Diseases

Shaden Kamhawi

co-Editor-in-Chief

Paul Brindley

co-Editor-in-Chief

 **Journal Requirements:**

1) We do not publish any copyright or trademark symbols that usually accompany proprietary names, eg ©,  ®, or TM  (e.g. next to drug or reagent names). Therefore please remove all instances of trademark/copyright symbols throughout the text, including:

- ® on page: 8.

Potential Copyright Issues:

i) Figure 1. Please (a) provide a direct link to the base layer of the map (i.e., the country or region border shape) and ensure this is also included in the figure legend; and (b) provide a link to the terms of use / license information for the base layer image or shapefile. We cannot publish proprietary or copyrighted maps (e.g. Google Maps, Mapquest) and the terms of use for your map base layer must be compatible with our CC BY 4.0 license.

5) In the online submission form, you indicated that The datasets used in this study are available from the corresponding author on reasonable request.. All PLOS journals now require all data underlying the findings described in their manuscript to be freely available to other researchers, either

1. In a public repository

2. Within the manuscript itself

3. Uploaded as supplementary information.

6) Please provide a detailed Financial Disclosure statement. This is published with the article. It must therefore be completed in full sentences and contain the exact wording you wish to be published.

1) Please clarify all sources of financial support for your study. List the grants, grant numbers, and organizations that funded your study, including funding received from your institution. Please note that suppliers of material support, including research materials, should be recognized in the Acknowledgements section rather than in the Financial Disclosure

2) State the initials, alongside each funding source, of each author to receive each grant. For example: "This work was supported by the National Institutes of Health (####### to AM; ###### to CJ) and the National Science Foundation (###### to AM)."

3) State what role the funders took in the study. If the funders had no role in your study, please state: "The funders had no role in study design, data collection and analysis, decision to publish, or preparation of the manuscript."

4) If any authors received a salary from any of your funders, please state which authors and which funders..

7) Your current Financial Disclosure states, "The author(s) received no specific funding for this work.".

However, your funding information on the submission form indicates receiving funds.

Please indicate by return email the full and correct funding information for your study and confirm the order in which funding contributions should appear. Please be sure to indicate whether the funders played any role in the study design, data collection and analysis, decision to publish, or preparation of the manuscript.

**Reviewers' comments:** Reviewer's Responses to Questions

**Key Review Criteria Required for Acceptance?**

**Methods**

-Are the objectives of the study clearly articulated with a clear testable hypothesis stated?

-Is the study design appropriate to address the stated objectives?

-Is the population clearly described and appropriate for the hypothesis being tested?

-Is the sample size sufficient to ensure adequate power to address the hypothesis being tested?

-Were correct statistical analysis used to support conclusions?

-Are there concerns about ethical or regulatory requirements being met?

Reviewer #1: -Are the objectives of the study clearly articulated with a clear testable hypothesis stated? YES

-Is the study design appropriate to address the stated objectives? YES

-Is the population clearly described and appropriate for the hypothesis being tested? YES

-Is the sample size sufficient to ensure adequate power to address the hypothesis being tested? YES

-Were correct statistical analysis used to support conclusions? Beyond my expertise.

-Are there concerns about ethical or regulatory requirements being met? NO.

Reviewer #2: The objectives of the study are clear, the study has a very robust sample size, the population is well described and appropriate, and the statistics are rigorous. However, I do have the following suggestions/concerns:

With regard to serology, the authors note that sampled individuals have similar demographic characteristics to dengue cases, but do they have similar geographic distribution?

Figure 1: Panel B is not legible and quite possibly not necessary.

Fig S2 does not instill much confidence in the association between elevation and temperature, given the scant number of temperature measurements above 500 m. Although temperature is certainly known to decrease with increasing elevation, is there an alternative source of more densely sampled data to better reveal this association in the Philippines? I do not think the figure is a deal breaker for the study, but its limitations should certainly be discussed if alternative data is not available.

**Results**

-Does the analysis presented match the analysis plan?

-Are the results clearly and completely presented?

-Are the figures (Tables, Images) of sufficient quality for clarity?

Reviewer #1: -Does the analysis presented match the analysis plan? YES

-Are the results clearly and completely presented? YES

-Are the figures (Tables, Images) of sufficient quality for clarity? YES

Reviewer #2: The analysis is robust, the results are clearly presented (though see some suggestions in minor critiques) and the figures are clear.

**Conclusions**

-Are the conclusions supported by the data presented?

-Are the limitations of analysis clearly described?

-Do the authors discuss how these data can be helpful to advance our understanding of the topic under study?

-Is public health relevance addressed?

Reviewer #1: -Are the conclusions supported by the data presented? YES

-Are the limitations of analysis clearly described? YES

-Do the authors discuss how these data can be helpful to advance our understanding of the topic under study? YES

-Is public health relevance addressed? YES

Reviewer #2: Generally the conclusions are well supported, the limitations are discussed, and the public health relevance is well articulated. I do, however, suggest the following changes:

1. Overall

It is not necessary to frame population density as a proxy for urbanization- population density is population density and important in its own right. Furthermore, population density may sometime be decoupled from other important markers of urbanization, such as NDBI. And lastly, the authors toggle back and forth between referring to population density and urbanization, even in the abstract, which is quite confusing. I recommend that urbanization be struck throughout the manuscript and replaced with population density.

2. Abstract

Delete second sentence; it is neither necessary nor entirely correct

**Editorial and Data Presentation Modifications?**

Reviewer #1: Please see summary and general comments.

Reviewer #2: Abstract

Change first sentence to

“Climate change is enabling geographic expansion of Aedes mosquito vectors into new areas.

Line 78: Add “than incidence” after “reliable metric”

Introduction

Rephrase the second sentence for correctness- not all Aedes species serve as dengue vectors, nor are all Aedes species found in warm, humid climates. Furthermore, even among the major dengue vectors, Ae. albopictus is more common at urban edges than in urban habitats proper.

Suggest you consider this paper, which shows higher Ae. aegypti abundance in rural than urban land covers:

Higher abundance of the vector Aedes aegypti in rural areas than in urban areas in Managua, Nicaragua.

Laguna HS, Díaz JM, Lopez MM, Balmaseda A, Harris E, Coloma J, Juarez JG.Res Sq [Preprint]. 2025 Mar 12:rs.3.rs-6059011. doi: 10.21203/rs.3.rs-6059011/v1.

Results

Figure 2F- need to specify that age is in years

Paragraph starting on line 284: First sentence of the paragraph contradicts the last sentence of the paragraph.

Figure 2D – reversing the orientation of elevation (high to low from left to right) relative to the panel above it (low to high from left to right) is problematic and will lead to misinterpretation- strongly suggest the x-axis be flipped to read from low to high.

Discussion

Line 387: Add FOI at the start of the sentence

Line 398: The sentence that starts on this line is a run-on; also “later” needs to be amended to “latter”.

**Summary and General Comments**

Reviewer #1: The authors compare the impact of altitude and population density on dengue incidence and force of infection across the Philippines, concluding that FOI is a more reliable metric for assessing transmission intensity than approaches that depend upon accurate clinical case reporting.

The manuscript is clearly presented but would benefit from additional references. I have highlighted some examples below, but I encourage the authors to review the text to ensure that key statements are appropriately supported.

The authors use the terms ‘altitude’ and ‘elevation’ interchangeably in the text, tables, and figures throughout the manuscript and supplementary materials. They may wish to settle on one of these terms. McVicar et al. provide a nice concise definition of each in the following manuscript: DOI 10.1007/s00442-012-2416-7

If permitted, for reasons of transparency and ease of access, I encourage the authors to share their data either as supplementary material or via a data repository. Otherwise, I only have a few small comments and suggestions for the authors.

Short title

Line 4: I suggest using key terms that agree with the title i.e., ‘population density’ rather than ‘urbanisation’, which has a slightly different meaning (something that is mentioned in the opening sentences of the abstract).

Abstract

Lines 78 – 79: For clarity, I suggest changing the text to “FOI provides a more reliable metric than incidence for understanding transmission dynamics and guiding interventions”.

Introduction

Line 122: Please include an appropriate reference at the end of the first sentence.

Lines 123 – 124: Not all Aedes species mosquitoes transmit dengue, so perhaps change to “Certain Aedes species mosquitoes, which transmit the dengue virus…” and include a suitable reference. In a word or two, you could also be clearer about the location of the warm, humid, urban environments to which you are referring e.g., the tropics, subtropics, or elsewhere.

Lines 133 – 134: As it is written, it seems like population densities are discrete units that need to be in proximity for transmission to occur. It may be more accurate to state 'High human population densities facilitate sustained dengue transmission between vectors and human hosts'.

Line 137 - 139: Please change “vectorial competence” to “vector competence”, which is the more commonly used term, and include a citation at the end of this sentence (that is assuming that you are referring to vector competence and not vectorial capacity – please do not confuse these terms, which might be worth briefly defining). Please also clarify the “daily fluctuations” to which you are referring in the final part of the sentence e.g., average/mean temperature, temperature range, or something else.

Methods

Line 188 – Please include a reference to the WHO 2008 criteria at the end of this sentence. I think it would also be beneficial if you briefly explained the meaning of ‘dengue warning signs’, since the term is used throughout the manuscript.

Results

Lines 261 – 263: The opening sentence of the results is slightly convoluted and could be refined. You mentioned the years of collection in the methods so you might consider excluding this information, and it is clear at this stage that the study is being conducted in the Philippines, so it doesn’t need to be mentioned twice in this sentence.

Lines 272 – 273: Is this statistically significant? It appears that only two of the weather stations were above about 250 m elevation in Figure S2.

Line 338 – 339: Since this subheading is “The combined impact…” you might wish to change the previous subheading to “The individual impact…” (or something appropriate) to more clearly define the separate analyses.

Discussion

Line 391: You may wish to amend the text slightly to highlight that ‘resources’ are often limited for combatting VBDs e.g., “to allocate limited resources to control interventions”.

Line 399: Please refer specifically to relevant vectors (e.g., Ae. aegypti/albopictus) here, since other Aedes species are more tolerant of cooler temperatures, though they aren’t necessarily vectors. Also change “later” to “latter”.

Figures

Fig 1. You use elevation/altitude interchangeably in the map and caption. Please be consistent in use of terminology.

Fig 1. The numbers and abbreviations in panel B are not defined in the caption (although some are later mentioned in the text). I suggest also defining them in the figure caption.

Fig 2. I noticed you flipped the x-axis for ‘Elevation’ in Panel D, and I understand that you have done this to show a left to right increase in exposure to DENV IgG. However, it is easy to miss. I suggest either maintaining the same direction (low to high elevation) shown in the other panels for elevation OR mentioning the change in direction/order in the figure caption.

Fig 3. A small legend for the colors in panels A and B would be nice. There is space for this, and it would help with quick interpretation of the figure.

Reviewer #2: This study interrogates the association of DENV transmission with both population density and elevation (a proxy for temperature) in the Philippines. The results reveal that both factors influence force of infection, but that these effects are not apparent when analyzing passively reported DENV cases. Furthermore, the study finds that the average age of dengue with warning signs cases correlates well with FOI, offering a more economical measure to track DENV transmission. The study is valuable in its country-wide scope, thorough analysis of FOI and incidence, and findings with translational relevance for dengue monitoring.

PLOS authors have the option to publish the peer review history of their article (what does this mean?). If published, this will include your full peer review and any attached files.

**Do you want your identity to be public for this peer review?** For information about this choice, including consent withdrawal, please see our Privacy Policy.

Reviewer #1: No

Reviewer #2: No

**Figure resubmission:** While revising your submission, we strongly recommend that you use PLOS’s NAAS tool (https://ngplosjournals.pagemajik.ai/artanalysis) to test your figure files. NAAS can convert your figure files to the TIFF file type and meet basic requirements (such as print size, resolution), or provide you with a report on issues that do not meet our requirements and that NAAS cannot fix.

After uploading your figures to PLOS’s NAAS tool - https://ngplosjournals.pagemajik.ai/artanalysis, NAAS will process the files provided and display the results in the "Uploaded Files" section of the page as the processing is complete. If the uploaded figures meet our requirements (or NAAS is able to fix the files to meet our requirements), the figure will be marked as "fixed" above. If NAAS is unable to fix the files, a red "failed" label will appear above. When NAAS has confirmed that the figure files meet our requirements, please download the file via the download option, and include these NAAS processed figure files when submitting your revised manuscript. **Reproducibility:** To enhance the reproducibility of your results, we recommend that authors of applicable studies deposit laboratory protocols in protocols.io, where a protocol can be assigned its own identifier (DOI) such that it can be cited independently in the future. Additionally, PLOS ONE offers an option to publish peer-reviewed clinical study protocols. Read more information on sharing protocols at https://plos.org/protocols?utm_medium=editorial-email&utm_source=authorletters&utm_campaign=protocols

---

## [Decision Letter · Decision Letter 1]

8 May 2026

Dear Dr Biggs,

We are pleased to inform you that your manuscript 'The impact of elevation and population density on dengue incidence and force of infection across the Philippines: Implications for climate-adapted surveillance' has been provisionally accepted for publication in PLOS Neglected Tropical Diseases.

Best regards,

Nikos Vasilakis

Section Editor

Nikos Vasilakis

Section Editor

Shaden Kamhawi

co-Editor-in-Chief

Paul Brindley

co-Editor-in-Chief

Reviewer's Responses to Questions

**Key Review Criteria Required for Acceptance?**

**Methods**

-Are the objectives of the study clearly articulated with a clear testable hypothesis stated?

-Is the study design appropriate to address the stated objectives?

-Is the population clearly described and appropriate for the hypothesis being tested?

-Is the sample size sufficient to ensure adequate power to address the hypothesis being tested?

-Were correct statistical analysis used to support conclusions?

-Are there concerns about ethical or regulatory requirements being met?

Reviewer #1: The authors have clarified their reasons for not submitting the data and have provided appropriate contact details for respective institutions. They otherwise satisfy the above criteria.

Reviewer #2: Authors have done an excellent job at responding to all critiques

**Results**

-Does the analysis presented match the analysis plan?

-Are the results clearly and completely presented?

-Are the figures (Tables, Images) of sufficient quality for clarity?

Reviewer #1: The quality of the figures is very good and the results are clearly presented throughout.

Reviewer #2: Authors have done an excellent job at responding to all critiques

**Conclusions**

-Are the conclusions supported by the data presented?

-Are the limitations of analysis clearly described?

-Do the authors discuss how these data can be helpful to advance our understanding of the topic under study?

-Is public health relevance addressed?

Reviewer #1: The authors address limitations of their study and otherwise provide a meaningful discussion. The public health relevance is clearly addressed throughout.

Reviewer #2: Authors have done an excellent job at responding to all critiques

**Editorial and Data Presentation Modifications?**

Reviewer #1: All relevant modifications have been addressed.

Reviewer #2: (No Response)

**Summary and General Comments**

Reviewer #1: The authors have carefully and thoroughly addressed my comments and have submitted a good revision of their manuscript.

Reviewer #2: Authors have done an excellent job at responding to all critiques

PLOS authors have the option to publish the peer review history of their article (what does this mean?). If published, this will include your full peer review and any attached files.

Reviewer #1: No

Reviewer #2: No

---

## [Editor Report · Acceptance letter]

Dear Dr Biggs,

We are delighted to inform you that your manuscript, "The impact of elevation and population density on dengue incidence and force of infection across the Philippines: Implications for climate-adapted surveillance," has been formally accepted for publication in PLOS Neglected Tropical Diseases.

Best regards,

Shaden Kamhawi

co-Editor-in-Chief

Paul Brindley

co-Editor-in-Chief
